**Data Availability Statement:** Samples of the transcripts are provided in the paper and Supporting Information files. Given the sensitive

# "This is you teaching you:" Exploring providers' perspectives on experiential learning and enhancing patient safety and outcomes in ketamine-assisted therapy

Elena Argento[1,2‡]*, Tashia Petker[1,2‡], Jayesh Vig[3☯], Cosette Robertson[1☯], Alexandria Jaeger[2☯], Candace Necyk[2,3☯], Paul Thielking[2☯], Zach Walsh[1☯]

**1** Department of Psychology, University of British Columbia, Kelowna, British Columbia, Canada, **2** Numinus Wellness Inc., Vancouver, British Columbia, Canada, **3** Faculty of Pharmacy and Pharmaceutical Sciences, University of Alberta, Edmonton, Alberta, Canada

☯ These authors contributed equally to this work.
‡ EA and TP are joint senior authors on this work.
* elena.argento@ubc.ca

## Abstract

### Objectives

This study explores therapists' perspectives on experiential learning, competencies, and training in ketamine-assisted therapy (KAT), a form of psychedelic-assisted therapy (PAT). We aim to understand how therapists' personal psychedelic experiences influence their self-perceived competency and therapeutic relationships regarding KAT.

### Methods

Licensed therapists from Numinus Wellness clinics in Canada and the USA who were trained in KAT were invited to participate in the study. Participation included a 60–90-minute semi-structured interview conducted remotely via secure videoconferencing. The interviews focused on the professional and personal impacts of providing KAT, its mechanisms, and the role of therapists' personal psychedelic experiences in delivering KAT. Data analysis utilized Interpretative Phenomenological Analysis (IPA) and a mix of deductive and inductive coding with Nvivo software.

### Results

Eight therapists (62.5% female, 37.5% male) were interviewed. All had formal training in KAT, with many also trained in other forms of PAT. All respondents endorsed the value of personal psychedelic experience for deepening understanding of clients' experiences and strengthening the therapeutic alliance. They all also expressed a desire for formal experiential training in KAT, which they viewed as a missing element in their training. Additional themes identified included the importance of relational safety and the therapeutic container, KAT's impact on professional development, competency, and purpose, and navigating risks and challenges in KAT, particularly with complex trauma clients.

nature of content and the detailed narrative structure of each interview, whole transcripts cannot be completely de-identified to be shared publicly. This is to protect the participants in the study. Queries regarding the data may be directed to the Advarra IRB at 905-727-7989 or https://www.advarra.com/contact-us/.

**Funding:** This study was supported by Numinus Wellness Research. The study funder had no role in the study design, data collection, analysis, interpretation, or writing of the manuscript. EA was supported by Canadian Institutes of Health Research (CIHR) [grant number: 164674] and Mitacs Elevate [grant number: IT34808 postdoctoral awards.

**Competing interests:** At the time of writing this manuscript, authors EA and TP were part-time consultants to Numinus Wellness Inc., and authors AJ, CN, and PT were employed by Numinus Wellness Inc. ZW is in paid advisory relationships with Numinus Wellness and Entheo Tech Biomedical regarding the medical development of psychedelic medicines and is a member of the Advisory Board of the Multidisciplinary Association for Psychedelic Studies (MAPS) Canada and MycoMedica Life Sciences. This does not alter our adherence to PLOS ONE policies on sharing data and materials.

## Conclusions

Our findings highlight the need for enhanced therapist training and evidence-based standardization of PAT programs that incorporate experiential learning. Such training has the potential to optimize safety and therapeutic outcomes.

## Introduction

Psychedelic-assisted therapy (PAT) is an approach that combines a psychedelic substance with a psychotherapeutic intervention, typically involving administration of a "classic psychedelic" (e.g., LSD or psilocybin) or psychedelic-like compounds (e.g., ketamine or MDMA). The changes in perception and perspectives that these compounds give rise to are considered to have therapeutic benefits particularly when delivered within a safe and supportive environment [1]. The limitations of currently available treatments for mental health conditions have led to a resurgence of interest in PAT, accompanied by several clinical trials identifying the potential applications of this modality in modern psychiatry [2]. The use of PAT in modern psychiatry is relatively new; however, the progress of psychedelics as medicine is largely based upon the long history of use by Indigenous peoples for a wide range of spiritual and healing purposes [3]. At present, clinical and research applications consist of structured protocols that provide psychedelic dosing sessions in carefully monitored settings with supportive non-drug psychotherapeutic preparation and integration sessions before and after dosing sessions.

Ketamine is currently the only widely available psychedelic compound for use in PAT, as psilocybin- and MDMA-assisted therapy remain illegal outside of clinical research and government approval pathways (e.g., Canada's Special Access Program). Ketamine, an N-methyl-D-aspartate (NMDA) antagonist, has been used as an anesthetic since 1964 and has a well-established safety profile [4]. Its off-label use as an adjunct to psychotherapy was first identified in the 1970s [5], and it has since been studied in numerous phase 2 and 3 clinical trials for treating mental health and substance use disorders. Mounting evidence for ketamine as an antidepressant agent led to rapid progress in broader applications of ketamine and ketamine-assisted therapy (KAT) for psychiatric conditions [6]. This includes the US FDA-approved ketamine nasal spray, esketamine, for treatment-resistant depression. A recent systematic review identified strong, rapid effects of ketamine for acute suicidality and depressive symptoms, and further evidence signaling promise for substance use disorders, obsessive-compulsive disorders, anxiety, and traumatic stress disorders [7]. Despite a great deal of variability in ketamine dose and route, comparisons of different methods have not found considerable differences in efficacy [8, 9]. Paralleling this emergent research is an increase in public interest for legal access to PAT, resulting in an explosion of new clinics offering KAT across North America.

While ketamine has been used for decades as a stand-alone treatment, the benefit of combining ketamine with therapy remains under investigation. The escalating interest in PAT has led to a proliferation of PAT training programs, with a wide range of therapeutic orientations and approaches to training. There is an ongoing effort to establish guidelines and norms for this rapidly developing PAT training landscape. It has been proposed that PAT therapists should possess six core competencies: empathic abiding presence, trust enhancement, spiritual intelligence, knowledge of psychological & physiological effects of the medicines used, self-awareness and ethical integrity, and proficiency in complementary techniques [10]. Although these competencies appear to have face validity, they have not been systematically evaluated in the context of current clinical practice. Despite the growing need for trained providers, the infrastructure of specialized training programs or regulatory certification remains immature.

Should providers of PAT have first-hand experience with psychedelics? Limited research suggests that providers' personal experience with psychedelics may improve outcomes for patients and that experiential learning may be an important component of training qualified therapists to deliver PAT [11–13], but evidence for both sides of the argument exist. The contents of forum discussions on "trip sitting" suggest that guides who had their own experience with psychedelics, and were knowledgeable about health and medicine, were strongly preferred by patients [14]. Direct experience with psilocybin has been rated as more important in providers of PAT than other individual characteristics, including the therapist's personal mental health experiences, experiences as a therapy client, gender, and ethnicity [15]. However, personal experience with psychedelics may also elicit negative judgments about therapists' professionalism, personality, respect for laws, or their personal relationship with substances [12, 16]. Notably, analysis of data collected from participants of the Spring Grove study, where 100 therapists used LSD for experiential training from 1969–1974, found that therapists rated the experience highly on educational value and understanding of psychedelics [11]. More recently, an FDA-approved phase 1 randomized controlled trial (MT1) aimed to expand the knowledge of trainees learning to conduct MDMA-assisted therapy [17]. Unpublished survey data gathered from 79 of 82 MT1 participants indicated substantial potential for professional and personal benefit and minimal risk of harm. Further research is warranted to understand how therapists' personal experiences with psychedelics affect their competencies in delivering PAT and how it may impact the therapeutic relationship and patient outcomes [18].

Given the gaps in knowledge raised above, the present study aimed to explore the perspectives of therapists delivering KAT regarding experiential learning, therapist competencies and training, and factors that contribute to enhancing safety and outcomes for patients. Through in-depth interviews with KAT providers, this qualitative study aims to inform best practices for training of KAT therapists and optimization of patient care.

## Methods

The present analysis draws on data from the Ketamine-assisted Therapy for Mental Health Conditions: A Qualitative Exploration to Support Evidence in Clinical Therapy (K-QUEST) study. K-QUEST is a mixed methods study that aimed to describe the lived experiences of clients and providers of KAT at Numinus Wellness clinics located in Canada and the USA. The study followed ethical guidelines regarding professional conduct, Good Clinical Practice in research, and confidentiality. The study holds ethics approval through Advarra's Institutional Review Board (CR00441515) and all participants provided written informed consent.

### Participants

Therapists eligible to participate in this study were individuals who have been trained to deliver KAT and are licensed healthcare providers (Registered Clinical Counsellors, Social Workers, Nurse Practitioners, and Physician Assistants) who have experience working with clients with mental health disorders. Therapists who had facilitated KAT sessions for at least two clients were recruited through Numinus Wellness clinics and invited to participate in a semi-structured interview. Participants received an honorarium in the form of a $25 gift card for volunteering their time.

### Interviews

Interviews followed an open-ended interview guide (see Supporting Information file) which aimed to explore the professional and personal impacts of facilitating KAT, facilitate deeper understanding of KAT's mechanisms of action, identify opportunities to improve the delivery

of KAT, and explore the role of therapists' direct personal experiences with ketamine or other psychedelics in shaping their knowledge and qualifications to deliver KAT. Each interview was conducted by one of two members of the research team (EA and TP) and lasted between 60–90 minutes each. All interviews were conducted remotely over secure videoconferencing, recorded, and transcribed verbatim.

## Data analysis

Two members of the study team (JV and CR) reviewed each audio file and transcript to correct for any auto-transcription errors. Cleaned interview transcripts were imported into a qualitative data management and analysis software program (Nvivo). Data were coded using both deductive and inductive approaches and involved the development of a codebook comprised of categories derived from the interview guide and expanded to include emergent codes and themes following review of the transcripts. The codebook was independently applied to each interview transcript by two researchers (a primary and secondary coder; EA and TP) to establish inter-coder agreement. Emergent codes and themes were then identified, and subthemes established in greater detail.

This study utilized Interpretative Phenomenological Analysis (IPA) as the primary methodology. IPA facilitates understanding healthcare and health conditions from the perspective of the participant and enables a detailed examination of personal lived experience, including the meanings behind the experiences and how participants interpret their experiences [19]. IPA has been the primary methodology used in other qualitative studies investigating PAT.

## Results

Overall, eight therapists were included in the present analysis (5 female and 3 male). Six therapists practiced in Canada and two therapists in the USA. Participants had, at a minimum, a master's level education; three participants achieved PhD level education. All participants had experience facilitating KAT sessions for at least two clients. Therapists in Canada had experience leading between two and eight KAT client protocols (which consisted of 3 ketamine sessions and preparation and integration sessions), whereas therapists located in the USA, where KAT has been provided for longer, had experience facilitating 12–100 KAT sessions. All therapists had received formal training in KAT through Numinus, and the majority had received additional training in other PAT such as MDMA-assisted therapy (e.g., Multidisciplinary Association of Psychedelic Studies/MAPS), or psilocybin-assisted therapy (e.g., Usona, ATMA, Fluence).

All participants reported prior experience with psychedelics, except for one participant who had experience with altered states of consciousness via breathwork. These experiences spanned therapeutic, ceremonial, recreational, and informal experiential learning contexts. Participants reported using psilocybin (63%; $n = 5$), LSD (50%; $n = 4$), and MDMA (38%; $n = 3$). Three participants reported ever using ketamine (38%; $n = 3$). One participant had used mescaline (peyote) and one participant had used dimethyltryptamine (DMT) (13%; $n = 1$). No participants had received formal experiential training with any psychedelic given the current legal restrictions on psychedelics. Other forms of altered states of consciousness were reported through shamanic drumming, hypnosis, yoga, tantra, sweat lodges, lucid dreaming, and childbirth.

### Main findings

**Expressed value of personal experience.** Participants were asked about their own personal experiences with psychedelics and the ways in which these experiences may have informed their practice of delivering KAT. One of the most prominent themes to emerge from

the narratives was that personal experience with psychedelics is viewed as extremely valuable among therapists and helps to build rapport and therapeutic alliance with clients. All participants, regardless of their prior experiences with psychedelics, expressed that having personal experience allows the therapist to gain **a deeper understanding and insights into the client's experience**. The narratives suggested that personal experience helps to bridge the gap between having an intellectual understanding of psychedelic effects and a deeper knowledge or felt sense of the psychedelic experience, which may aide in the therapeutic process for clients:

*There was an understanding there that was. . .better absorbed, right, because I'd had my own experience with it. . . It was far easier after [my own experience], and it has remained this way, to understand more fully when a client was explaining their experience, when they were doing their best to build words around something that is otherwise very difficult to explain. (participant #2007)*

*I haven't had any experiences with psychedelics yet, which is something that I really want to experience because part of the rationale is that it's such a different state that you want to be able to understand it from an experiential standpoint. . . I don't know if I would say that it's essential, but I would say it's ideal and important. . . To know first-hand what it must be like for someone to be so open and vulnerable. (participant #2002)*

**The therapeutic alliance** is thought to be a fundamental aspect of creating a supportive set and setting in PAT and a key determinant of long-term positive outcomes [20, 21]. Along these lines, participants described that personal psychedelic experience helps the therapist to create an **enhanced sense of trust and safety**, which allow the client to go further in their experience:

*I don't feel like I would be able to counsel people very well if I don't have any idea or context of where they've been with these things. It's also really comforting to people when they're in those really unfamiliar states. . .like when they ask, "Do you know what this is like?" I can just say yes. I have no problem with that. . .I used to be really closed about it, but. . .I'm done being in that closet. . . It gives them the ability to let go more easily and to trust in the experience. . . If they know that there's someone there who's like, an anchor, and knows the terrain, then I think that they can get more out of the experience. (participant #2009)*

*To me, it comes to the ability to hold space. Like the emotional intensity when you've been there, and you know the person's going to push through and there's no danger. I think that's mostly the way I see it, like building trust in those states. . . My psychedelic use. . .brings me a sense of grounded-ness and of stability. (participant #2003)*

*I do think that there's something about earning trust or creating a safe space by saying I've done my work too. I've tried these substances. I have a lot of respect and humility for the power, and I know what to expect. . . If I were the client, I would feel safer with someone who had done it themselves. (participant #2001)*

Several therapists also noted that gaining a deeper understanding of these vulnerable and sometimes challenging states through personal experience **increases capacity for empathy**:

*Having had some of those experiences myself, I can really empathize with, you know, when people are struggling. . .like I can really relate to the struggle. And letting go of that and kind of surrendering yourself to the medicine. (participant #2005)*

*My actual participation with ketamine has opened up access to insights that were always there. They're just a lot more available now for me to be in touch with and to share. I think that comes out as an empathy, for the most part, with my clients. The feedback I get from them around that is quite positive. (participant #2007)*

**Desire for experiential training.**   Participants were asked specifically about experiential learning as part of training to deliver psychedelic-assisted therapies. While most (*n* = 7) participants had prior personal experience with psychedelics, including some with ketamine (*n* = 3), all participants (*n* = 8) expressed a desire for formal experiential learning and that this was a key element missing from their training to deliver KAT:

*I think that having an experience with the medicine as part of the training would be really important. I think in fact that is what was missing. . . I feel a little bit blind when I am working with the client and ketamine, having not had an experience. (participant #2001)*

*I've had clients also ask me, like, oh, have you had your own experience? And it feels not like unethical, but like it doesn't feel right to me that I can't say to them like I have a felt sense of what it is that you're experiencing. So, I'm really missing that. . .experience with the medicine. . .that's a real missing piece. (participant #2006)*

Many therapists expressed an **eagerness to undergo KAT for themselves**, especially if offered in a supportive, controlled setting. Participants expressed **frustration around barriers to safely accessing experiential training** due to regulatory, insurance or other organizational reasons. While some participants were not motivated to seek experiential training independently, at least three participants chose to engage informally in experiential training either underground with other colleagues or as patients in other KAT clinics to obtain the experience for themselves:

*[If there was] an opportunity for therapists to have a supported ketamine experience I would take them up on it, you know, in a container with a lot of safety. But I don't feel motivated to go out and do that on my own. . . I have done some underground work that I would say has been the most valuable source of learning. The theory is important, but you know, we don't know how to operationalize that until we actually see it in practice. (participant #2005)*

*I would really like the opportunity to try [ketamine] within a clinical or healing context. . . It really feels like a gap in my experiential knowledge. . . (participant #2006)*

*I'm doing [experiential training with ketamine] informally with colleagues now because it's missing in the training. . . that's how we're really going to learn a lot more than just the sort of lecture or whatever, that seems overly theoretical. (participant #2001)*

*I love the idea of [experiential learning with ketamine] being a formal, organized part of the actual training. . . The earlier the better, simply because that part of it informs so much of the practice. (participant #2007)*

Ketamine may offer unique experiences that cannot be fully understood by having experiences with other psychedelics. The narratives suggested that **experiential training with ketamine specifically would be beneficial**. The reasons for this included being able to understand the specific effects of different ketamine routes of administration and personal insights that better equip therapists to guide their clients by **enhancing therapists' intuitive understanding and overall confidence** in the modality:

*What would have prepared me is experiential training for sure on the ketamine, especially on the modality like the sublingual [formulation] and like just being able to experience it from the other perspective. (participant #2003)*

*[Personal experience with ketamine] provided a certain degree of insight and a more confident access to my own intuition, which I wanted on board for myself personally, but also in the work that I'm doing. In fact, those were specifically some of the intentions I had carried into my very first ketamine experiences. . . I wanted to be able to more comfortably access my own intuition, my own confidence, so that I can provide the kind of care and support for my patients, you know, that I would want for myself. (participant #2007)*

*I think that there will always be a limitation to knowledge. It will be hard to bridge that experience and I feel that way with ketamine at times where it's like I can tell you what it feels like to go into altered states of consciousness. I can connect to that. Some of the vulnerability and the uncertainty. But I can't tell you what this ketamine experience is going to feel like and certainly that does feel like a limitation for me. (participant #2005)*

**Relational safety & therapeutic container.** Relational safety—the importance of establishing a trusting, supportive therapeutic container–emerged throughout the narratives as a key theme. Participants discussed various mechanisms of action for KAT, including **helping clients access deep-seated trauma and develop self-compassion, ketamine being a catalyst for emotional breakthroughs, and empowering clients to connect with their inherent inner wisdom**. However, the relational safety piece was identified as foundational to creating an environment that facilitates the therapeutic experience leading to new insights and growth.

Relational safety was described as enabling clients to become more vulnerable, as they could feel more secure and better able to confront and integrate challenging or painful memories and emotions:

*When there's more of an alliance, a good, safe relationship that has been established, there's more ability to be vulnerable. . . The gist of the healing work is really just to get in touch with these sort of exiled places, these places that they've lost touch with and to reintegrate them into their system, into their consciousness. But to be able to do that, you have to go there and you have to experience something very uncomfortable. . . I think it's just crucial that there's a really solid alliance, like really solid trust that I can do anything with this therapist in front of me. (participant #2001)*

*Shaking, trembling, bawling, sobbing, heaving. You know, first of all, I don't think that would have ever happened if [the client] was alone, because they would have never felt safe enough to go there. (participant #2006)*

While the medicine is an important catalyst, relational safety and therapeutic alliance between the therapist and client was identified as **a critical prerequisite to facilitate a deeper exploration and healing process** through KAT:

*Relational safety needs to be really built before introducing the substance. . .because even with the medicine on board, our protective parts still do what they're meant to do and that relational safety will be the ingredient that actually gives people the opportunity to go deeper, right? It gives people enough safety in their own bodies to access that. (participant #2005)*

*Being able to build a therapeutic alliance that can be just as supportive as the medicine is. . .you know they're interdependent. I mean they need each other, they rely upon another, that alliance and the medicine. They both really need to be on board. (participant #2007)*

*I think that depending on the safety [the client will] develop, they will go deeper in their journey. (participant #2004)*

Along these lines, participants **reiterated the importance of the therapy component of KAT**:

*I don't know how you could take the therapy out of it and get the same positive results. (participant #2006)*

*I get so excited at the idea that one day [KAT] can be so much more driven by the therapy. We don't want this to be a medicine factory. (participant #2007)*

**Impact on professional development, competency, and purpose.**   Participants described that their experiences delivering KAT have led to a **deeper exploration of their roles as therapists**, evaluating their own expectations, desires, and professional ethics:

*[Being a KAT therapist] has brought up more questions, I think, than satisfaction or dissatisfaction. It's more like, What kind of therapist am I? What kind of therapist do I want to be? And how do I want to show up in the psychedelic space? (participant #2005)*

*Because psychedelics are such amplifiers and there's such a sensitivity to energy in the space. . .I feel even more committed to. . .ethical practice, that you really can't cut corners when it comes to this kind of work. (participant #2006)*

Some participants explained that their experiences as KAT therapists have led to a process of **self-reflection and growth** that involved a shift away from offering knowledge and advice and towards **empowering their clients to realize and mobilize their own inner wisdom and capabilities to heal**:

*Ketamine is not a mystical thing that goes in and introduces brand new knowledge or wisdom or anything like that. It's not installing experience in you. This isn't me, you know, showering my own wisdom upon you. This is you teaching you. The medicine goes in and allows you to access parts of you that are already very inherently and instinctively wise and productive and helpful. (participant #2007)*

*I think there's always that reminder to get out of the way. . .people have what they need inside them, right? Stop needing to make an outcome happen or make sure the client feels great when they leave or achieve symptom reduction quickly. Stop trying to direct the process so much and just meet people in that loving presence. . .so that they can access that in themselves. (participant #2005)*

A prominent theme to emerge from the narratives was that guiding people through KAT has generated a **deep sense of fulfillment, reward, and purpose** in the work therapists do. Witnessing profound and accelerated transformations in their clients was highlighted as an inspiring and motivating factor to continue this line of therapy:

*I get to witness a certain degree of healing right there in the moment, and that's, I think, of course why probably most therapists become therapists. They're looking to have that kind of fulfillment. They go home at the end of the day and they're exhausted, emotionally worn down, but there's this silver lining of, wow, you know, some really important things happened*

*today for some good people who are trying their hardest, and maybe for the first time in a long time, they're seeing results and they're hopeful, they're feeling empowered, they feel like there's a course forward, and they have new tools that they never even knew about before they walked in. (participant #2007)*

*There's something really inspiring about seeing people have breakthroughs that I know would take years in therapy. . . Regular therapy, even like cutting-edge trauma therapies, you know, would take forever. And so, I find that really inspiring and it makes me want to continue and see if we can make these treatments even more accessible to as many people as possible. . . There's a really deep rich sense of meaning and purpose that I have found in doing this work. (participant#2006)*

*That privilege to be able to accompany someone [in their healing journey] and that it's often very meaningful for people, and you get to be a part of that. (participant #2002)*

Some participants also highlighted the dual benefit of **boosting their own mental wellness, self-awareness, and spiritual development**, all of which are considered important traits or core competencies in providers of PAT [10]:

*Psychedelics were a real turning point for me. I feel incredibly passionate about it, not just professionally or clinically, but really personally, and not just my healing journey but my own spiritual development as a person and as a human being. (participant #2006)*

*It is such an inspiration for me to connect with people and to also transform myself through their own transformation, and to receive a lot from this connection. (participant #2004)*

**Navigating risks & challenges.**   As with all PAT, and therapy in general, KAT is not without risks. Participants highlighted some key safety concerns and shared their thoughts on ways to navigate or mitigate certain risks and challenges.

Participants discussed **potential adverse outcomes, such as increased or intensified symptoms**, including suicidality and hospitalization following ketamine experiences. The unpredictable nature of psychedelic experiences in general was highlighted, and at least 3 therapists spoke to the **profound responsibility when working with complex clients and people who have complex trauma**. One therapist in particular questioned KAT as best practice in a case where the client's symptoms got worse:

*[The client] was suicidal because he was like, "This didn't work. Now I have nothing left. There's no hope." He was in the hospital for 3 months after his second ketamine session. . . You have experiences like that and you think, hmmm, are we sure this is best practice? (participant #2006)*

*That first medicine session, it can be really scary, especially for people that never went through psychedelic experiences. . . and like, the backlash afterward is really difficult. Some people will feel worse and more depressed, more suicidal, and so this is a huge piece—to take care of and to prepare for. (participant #2004)*

*We're dealing with real people with real issues. . .you have to be careful with what we're doing. . .it's not magic. And so it [a difficult session with a client] shook me to my core, I gotta say, and especially what happened afterwards made me realize the need for more training. So I did do some training of about complex trauma afterwards. (participant #2003)*

**Managing client expectations and/or disappointments** emerged as a key theme from the narratives. Participants highlighted that psychedelics are not a magic pill and healing is multi-faceted. As such, there is a need to manage expectations, and some clients experience disappointment following KAT:

*Unfortunately, it's not an uncommon experience. . .for clients to come away actually really disappointed with their experiences. (participant #2005)*

*People are suffering and so they just want a pill that will heal them forever but. . .really highly traumatized clients and complex clients will not necessarily access that level right away or ever. . .and so I'm just a bit more realistic right now. (participant #2004)*

Participants also spoke to recognizing the various needs of individuals who arrive at KAT, their unique histories, and the impact of past traumatic experiences in previous medical settings. In this context, at least 4 participants expressed **the need for flexibility in the timing and number of sessions**, and again the importance of having a **strong therapeutic container**:

*I do think that it's difficult with two or three prep sessions that for most people, especially people who are coming in with treatment-resistant conditions, to have adequate trust in the therapist. . .And then when there's not enough trust, I don't think that they would surrender to the medicine sufficiently. . . [For example,] I had to get [the client] out of the experience to debrief. . .and he wasn't ready to come out of it. He was actually right in the middle of a very powerful experience. . . We didn't even have a chance to debrief about his experience. . .it requires a lot more flexibility I think to work with these substances. (participant #2001)*

*[If] there's not enough prep sessions. . .you don't have enough of an alliance for the person to be able to go through this very vulnerable experience and really surrender to it. . . Not all clients will be able to do a medicine session after they've only met with you twice, right? That's going to be hard for some of them. (participant #2002)*

*I love the idea of, hey, this is your space. Let's take some time getting ready for the medicine session, whatever that involves. Let's make sure that your experience with the medicine is as brief or as long as it needs to be, because we don't always get to predict just how long that's going to be. (participant #2007)*

## Discussion

The present study describes findings from in-depth interviews with providers of KAT, elucidating perspectives on the potential value of experiential learning and broader impacts on professional development, as well as enhancing safety and efficacy outcomes for patients. The narratives offer key insights into therapists' lived experiences with facilitating KAT and suggest ways in which competencies may be bolstered through experiential training, and the importance of building a strong yet flexible therapeutic container to navigate risks and challenges. This study builds upon the growing body of PAT literature and provides a starting point for understanding therapists' experiences and perspectives to help inform best practices for providing KAT. Overall, five main themes emerged from the analysis: 1) expressed value of personal experience with psychedelics, 2) desire for experiential training, 3) impact on professional development, competency, and purpose, 4) relational safety and therapeutic container, and 5) navigating risks and challenges.

Participants in our study expressed that personal experience with psychedelics led to a deeper exploration of their role as a therapist and facilitated professional and personal growth. Psychedelic experiences were perceived to improve their own mental wellness and spiritual development, and to enhance self-awareness about their role and relationship with clients. Personal experience also provided therapists with a deeper understanding of clients' experiences and instilled confidence in their ability to support clients through confronting and integrating challenging memories and emotions.

Our findings echo previous research exploring the potential benefits of PAT for non-clinical populations and therapy providers. In a study of MDMA-assisted psychotherapy, a therapist who has been on "both sides of the couch" (i.e., experienced psychedelics as a patient and therapist) declared their personal experience with psychedelics helped them to reflect and change their therapeutic practice in positive ways [22]. Similar sentiments were frequently reported by our participants. In a study that reported on findings from a survey of 32 therapists involved with psilocybin-assisted therapy for depression, 88% had personal experience with psychedelics, with personal development and spiritual growth being cited among motives for use [23]. Furthermore, KAT has been demonstrated to have mental health benefits for healthcare providers struggling with depression, anxiety, and PTSD [24]. The literature also highlights benefits among 'healthy' volunteers that extend beyond treating a mental health diagnosis (e.g., overall wellbeing, life satisfaction) [25, 26]. The current study adds to this work by providing integrated accounts of benefits to both personal and professional development.

While still under investigation, it has been reported that some patients view personal experience with the psychedelic substance as being important in a PAT therapist, with this view being held especially by people of color and those who had previously received therapy [15]. This is in line with the broader view that accommodating client preferences is crucial for therapy. According to a meta-analysis of over 16,000 clients across 53 studies, accommodating client preferences in psychotherapy improves treatment outcomes and client retention rates [27]. We can reasonably predict that accommodating client preferences for a therapist with personal psychedelic experience will improve client engagement, retention, and treatment response. There are, however, some cautions to consider with respect to experiential training. It has been argued that future research is needed to explore whether a therapist's or researcher's personal experience with psychedelics introduces bias to the field and a tendency to demonstrate excess enthusiasm for PAT, given the highly meaningful and transcendent states that psychedelics can induce [23, 28]. Ultimately, understanding the therapeutic alliance and alignment with patient values will be central to understanding best practices in PAT.

Our findings also align with extant literature underscoring the importance of establishing relational safety and a strong therapeutic alliance with PAT patients. These critical elements in terms of enhancing both safety and outcomes when delivering PAT appear to be bolstered by the provider's personal experience. Experiential knowledge both increases confidence in their ability to provide the therapy, and also strengthens the belief that the patient has their own inner wisdom and capacity for change [10, 13]. Studies suggest that experiential learning with psychedelics may therefore be an important adjunct to therapist training programs [12, 13, 22]. This notion is aligned with some other psychotherapy training programs, whereby providers are encouraged to have personal experience with the therapies being delivered [29]. While experiential training may not need to be an outright requirement, growing evidence suggests that there may be strong benefits to providing opportunities for experiential learning as part of PAT training programs.

Findings from this study speak to some of the risks and challenges of KAT, particularly when working with patients with complex trauma. The risk for adverse outcomes may be mitigated by enhancing therapist training, particularly around trauma-informed care, but also

building flexibility into KAT protocols to offer adaptable KAT session durations and more preparation and/or integration sessions for those who need it, which in turn strengthens relational safety and the therapeutic container. Additionally, research surrounding KAT implementation in patients with complex trauma will allow for the identification of factors crucial to accentuate positive treatment outcomes and mitigate challenges that may arise with such patient groups.

Finally, narratives in this study relayed a strong desire for experiential training with ketamine specifically, and frustrations around the limited options and regulatory hurdles to access. This is based on the notion that therapists' first-hand experience with psychedelics enhances their competency in delivering PAT. This notion was strongly supported by reports from our participants. However, existing opportunities for experiential training are challenging to obtain. Access is limited to clinical trials which are not readily available to most providers, medical use whereby ketamine is prescribed as an adjunct to therapy for those with a mental health diagnosis that meet eligibility criteria, and underground or illegal procurement which may not be safe or ethical. Past research coupled with this resounding narrative echoed by all participants in the present study suggests a need for increased legal access to psychedelics as formal experiential training options for those who work closely with these substances in professional settings.

## Strengths and limitations

There are several limitations to consider with the current study. The study included a small convenience sample which introduces the potential for self-selection bias and narratives were subject to recall bias. The sample was limited to Numinus KAT clinics in Canada and the USA, and findings may not be generalizable to other KAT providers. Further, there are significant cost barriers to accessing KAT which may limit therapists to experiences with client support and relational needs that may not represent the range of challenges among patients with fewer financial resources. Incorporation of KAT into existing mental health treatment frameworks, including alternative routes of access to therapy, will allow for a deeper understanding of the factors relevant to optimizing KAT and KAT provider training for diverse populations. As the current study is qualitative, we are unable to draw causal associations between therapist responses and reported outcomes in their clients. However, the qualitative approach employed an IPA methodology which allowed for a nuanced understanding of participant perspectives and elucidated their lived experiences, including how they make meaning of or interpret their experiences. It is possible that participants may have answered questions in a way they perceived as helpful or desirable; however, participants were assured that their confidentiality and anonymity was protected by an independent research team who collected and analyzed the data.

## Conclusions

Given the rapidly rising demand for ketamine and other psychedelic-assisted therapies, there remains an urgent need to enhance training for therapists. Evidence-based standardization of PAT training programs is a crucial step to ensure therapists have the necessary qualifications and skills to work with psychedelics in a manner which maximizes therapeutic outcomes in patients, while also minimizing any risk of adverse events. While more research is required, current findings suggest that experiential learning might be combined with specialized PAT training programs to strengthen professional competency and optimize safety and outcomes for patients and therapists. PAT is a distinct intervention requiring a specialized skillset. As it continues to gain traction in research and clinical settings, provider perspectives offer a valuable resource for the development of best practices.

## Supporting information

**S1 Appendix. Qualitative interview guide.**
(PDF)

## Acknowledgments

The authors gratefully acknowledge all those who contributed their time and expertise to the study, especially the research participants who shared their perspectives and stories.

## Author Contributions

**Conceptualization:** Elena Argento, Tashia Petker.

**Data curation:** Elena Argento, Tashia Petker, Jayesh Vig, Cosette Robertson, Alexandria Jaeger, Candace Necyk, Zach Walsh.

**Formal analysis:** Elena Argento, Tashia Petker.

**Methodology:** Elena Argento, Tashia Petker, Jayesh Vig, Cosette Robertson, Zach Walsh.

**Project administration:** Alexandria Jaeger, Candace Necyk.

**Supervision:** Elena Argento, Paul Thielking, Zach Walsh.

**Writing – original draft:** Elena Argento, Tashia Petker.

**Writing – review & editing:** Elena Argento, Tashia Petker, Jayesh Vig, Cosette Robertson, Alexandria Jaeger, Candace Necyk, Paul Thielking, Zach Walsh.

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
