## [Decision Letter · Decision Letter 0]

17 Mar 2024

PONE-D-23-41326“This is you teaching you:” Exploring providers’ perspectives on experiential learning and enhancing patient safety and outcomes in ketamine-assisted therapyPLOS ONE

Dear Dr. Argento,

Thank you for submitting your manuscript to PLOS ONE. After careful consideration, we feel that it has merit but does not fully meet PLOS ONE’s publication criteria as it currently stands. Therefore, we invite you to submit a revised version of the manuscript that addresses the points raised during the review process.

We look forward to receiving your revised manuscript.

Kind regards,

Andrea Mastinu

Academic Editor

PLOS ONE

[At the time of writing this manuscript, authors EA and TP were part-time consultants to Numinus Wellness Inc., and authors AJ, CN, and PT were employed by Numinus Wellness Inc. ZW is in paid advisory relationships with Numinus Wellness and Entheo Tech Biomedical regarding the medical development of psychedelic medicines and is a member of the Advisory Board of the Multidisciplinary Association for Psychedelic Studies (MAPS) Canada and MycoMedica Life Sciences.]. 

Reviewers' comments:

Reviewer's Responses to Questions

**Comments to the Author**

1. Is the manuscript technically sound, and do the data support the conclusions?

Reviewer #1: Yes

Reviewer #2: Yes

2. Has the statistical analysis been performed appropriately and rigorously? 

Reviewer #1: N/A

Reviewer #2: Yes

3. Have the authors made all data underlying the findings in their manuscript fully available?

Reviewer #1: Yes

Reviewer #2: Yes

4. Is the manuscript presented in an intelligible fashion and written in standard English?

Reviewer #1: Yes

Reviewer #2: Yes

5. Review Comments to the Author

Reviewer #1: The main claims of the paper revolve around the necessity for enhanced therapist training and standardized programs in ketamine-assisted therapy (KAT), drawing insights from therapists' perspectives. These claims hold significant importance for the discipline of psychedelic-assisted therapy as they address crucial aspects such as experiential learning, competencies, and training, which are pivotal for optimizing therapeutic outcomes and ensuring patient safety. By highlighting the value of personal psychedelic experiences in therapist education and emphasizing the need for formal experiential training, the paper contributes to shaping best practices in the field.

In terms of contextualization within the previous literature, the authors appropriately position their claims by acknowledging the emerging interest in KAT and the broader landscape of psychedelic-assisted therapy. They engage with existing literature by discussing the potential therapeutic benefits of KAT and the gaps in therapist training and standardization, demonstrating a fair treatment of the literature.

In terms of organization and clarity, the manuscript is generally well-structured and accessible, making it suitable for both specialists and non-specialists in the field. However there are some modifications that the authors should make:

1) The experimental plan used should be better explained

2) How was Ketamine administered?

3) Ketamine dosages have not been reported.

4) How were the dosages of the study chosen considering the fact that therapists are in health? The dosage/route of administration used for the therapists were in the range normally used for their patients?

5) There was a correspondence in the dosage as long with the route of administration among all therapists (Canadian and American) ?

6) Did they observe differences between the experiences proved by female and male therapists?

I understand that these observations could be exempt from the objectives of the paper as a qualitative analysis was reported, however, I think that these parameters should be included to give strength to the findings and provide a more comprehensive understanding of therapists' perspectives.

Reviewer #2: I recommend expanding your sample size. I believe this sample is too small (especially given the relationship to the clinic) to contribute meaningful findings to the literature. See https://www.sciencedirect.com/science/article/pii/S0277953621008558

and the recommendation that samples of 9-17 were needed to reach saturation.

6. PLOS authors have the option to publish the peer review history of their article (what does this mean?). If published, this will include your full peer review and any attached files.

Reviewer #1: No

Reviewer #2: No

---

## [Author Response · Author response to Decision Letter 0]

28 May 2024

Comments Regarding Journal Requirements:

Comment 1: When submitting your revision, we need you to address these additional requirements.

Author Response: 

Thank you for pointing this out. We have updated the headings throughout the paper to match the journal’s instructions. We have also updated the affiliations section of the manuscript on Page 1, including the corresponding author section. The supporting information (qualitative interview guide) has been removed from the manuscript and uploaded as a separate file as per the guidelines.

Comment 2: We note that the grant information you provided in the ‘Funding Information’ and ‘Financial Disclosure’ sections do not match. 

Author Response: We have reviewed what was originally submitted and can confirm that the following statement is correct regarding Funding Information and Financial Disclosure: 

This study was supported by Numinus Wellness Research. The study funder had no role in the study design, data collection, analysis, interpretation, or writing of the manuscript. EA was supported by Canadian Institutes of Health Research (CIHR) and Mitacs Elevate postdoctoral awards.

It is not clear what information did not match and we are unsure where we can update this on the revision submission site. We did, however, update the financial disclosure statement on the first page of the manuscript with the grant numbers as requested. We also added Numinus Wellness Inc. as a funder under "Funding Information"

on the revision submission page. There was no specific grant number or recipient for this funding as the company allocated overall operational funds to the research department. The funding was not project specific. If the editorial team requires further editing to these sections, please let us know and we are happy to work with someone to fulfill any additional requirements.

Comment 3: Thank you for stating the following in the Competing Interests section: 

[At the time of writing this manuscript, authors EA and TP were part-time consultants to Numinus Wellness Inc., and authors AJ, CN, and PT were employed by Numinus Wellness Inc. ZW is in paid advisory relationships with Numinus Wellness and Entheo Tech Biomedical regarding the medical development of psychedelic medicines and is a member of the Advisory Board of the Multidisciplinary Association for Psychedelic Studies (MAPS) Canada and MycoMedica Life Sciences.]. 

Author Response: 

Thank you for clarifying this. We have updated this statement on Page 1 of the manuscript as requested and confirm that this does not alter our adherence to all PLOS ONE policies on sharing data and materials. We have included an updated competing interest statement in our Response to Reviewers cover letter with this submission. 

Comment 4: Please include your full ethics statement in the ‘Methods’ section of your manuscript file. In your statement, please include the full name of the IRB or ethics committee who approved or waived your study, as well as whether or not you obtained informed written or verbal consent. If consent was waived for your study, please include this information in your statement as well. 

Author Response: 

On Lines 173-176, the manuscript was edited to include the full ethics statement in the Methods section of the paper under “Participants”. It now reads (with the full ethics statement bolded below for clarity):

The study followed ethical guidelines regarding professional conduct, Good Clinical Practice in research, and confidentiality. The study holds ethics approval through Advarra’s Institutional Review Board (CR00441515) and all participants provided written informed consent. Participants received an honorarium in the form of a $25 gift card for volunteering their time.

We also shifted the first and second last sentences of the Methods section from the subheading “Participants” to above this subheading to better reflect overall Methods. This is clearly detailed in the tracked changes.

Comment 5: Please include captions for your Supporting Information files at the end of your manuscript, and update any in-text citations to match accordingly. Please see our Supporting Information guidelines for more information: http://journals.plos.org/plosone/s/supporting-information. 

Author Response: 

Thank you. This has been corrected, with a caption added and the file removed from the manuscript and uploaded separately.

Reviewer 1 Comments:

The main claims of the paper revolve around the necessity for enhanced therapist training and standardized programs in ketamine-assisted therapy (KAT), drawing insights from therapists' perspectives. These claims hold significant importance for the discipline of psychedelic-assisted therapy as they address crucial aspects such as experiential learning, competencies, and training, which are pivotal for optimizing therapeutic outcomes and ensuring patient safety. By highlighting the value of personal psychedelic experiences in therapist education and emphasizing the need for formal experiential training, the paper contributes to shaping best practices in the field.

In terms of contextualization within the previous literature, the authors appropriately position their claims by acknowledging the emerging interest in KAT and the broader landscape of psychedelic-assisted therapy. They engage with existing literature by discussing the potential therapeutic benefits of KAT and the gaps in therapist training and standardization, demonstrating a fair treatment of the literature.

Reviewer Question 1: In terms of organization and clarity, the manuscript is generally well-structured and accessible, making it suitable for both specialists and non-specialists in the field. However there are some modifications that the authors should make:

1) The experimental plan used should be better explained

2) How was Ketamine administered?

3) Ketamine dosages have not been reported.

4) How were the dosages of the study chosen considering the fact that therapists are in health? The dosage/route of administration used for the therapists were in the range normally used for their patients?

5) There was a correspondence in the dosage as long with the route of administration among all therapists (Canadian and American) ?

6) Did they observe differences between the experiences proved by female and male therapists?

I understand that these observations could be exempt from the objectives of the paper as a qualitative analysis was reported, however, I think that these parameters should be included to give strength to the findings and provide a more comprehensive understanding of therapists' perspectives.

Author Response: 

We thank this reviewer for their feedback and positive comments regarding the structure and accessibility of our manuscript. To be clear, our study was a qualitative exploration that involved interviews with therapists recruited through KAT clinics and thus did not involve any administration of ketamine to therapists or clients. Administration of ketamine to clients of the clinics was outside the scope of our study; the aim of the present study was to explore perspectives of KAT therapists via qualitative interviews regarding their experiences with providing KAT with their clients. As such, there was no experimental intervention. Please see text included in the Methods section for more details. A primary finding in our study was that none of the therapists received ketamine as part of their formal training in KAT, and all felt this was a major limitation in their training having not had direct experience with that medicine. 

We did not examine gender differences in therapists’ interview responses because this was not an a-priori research question and is beyond the scope of the present study’s aims. Given the very limited sample to draw from, meaningful conclusions cannot be made by further partialling the already small sample size of 8 into comparison groups based on gender. We greatly appreciate the detailed thoughts from this reviewer and agree that any future study on the administration of ketamine in therapists should include all of the methodological considerations that were recommended. 

Reviewer 2 Comments:

Reviewer Question 1: I recommend expanding your sample size. I believe this sample is too small (especially given the relationship to the clinic) to contribute meaningful findings to the literature. See https://www.sciencedirect.com/science/article/pii/S0277953621008558

and the recommendation that samples of 9-17 were needed to reach saturation.

Author Response: 

We thank this reviewer for raising their concerns regarding sample size. However, we feel strongly that our findings remain worthy of publication and would like to provide further justification and context for your consideration.

Publishing qualitative research with a small number of participants remains valuable when the insights are novel, the context is unique, and/or the findings contribute significantly to a specific field. Given that this is the first study examining therapists’ first-hand perspectives on practices of KAT, our findings are indeed novel and expected to contribute significantly to a new line of inquiry important for both research and clinical practice. Given the paucity of literature on therapist competencies and experiences in practice of KAT and psychedelic-assisted therapy more broadly, we believe our manuscript contributes meaningfully to the growing field of psychedelic therapy and medicine. It has been indicated in the manuscript that these findings should serve as a starting point for future research. 

The sample size of 8 (only one less than the recommended minimum of 9 cited above) is justified by consideration of practical restrictions, the nature of the study population, and quality of data obtained. This project was completed with constraints for both time and funding, and as such we are not able to collect more data as the study has ended. Future qualitative research with more participants would certainly strengthen findings and we hope our study encourages further investigation of therapists’ perspectives. Notably, we were able to obtain data with a level of specificity and depth to provide rich and valuable contextual information and several of our themes extracted were near-unanimous sentiments among the therapists interviewed. 

We agree with this reviewer that the small sample size is a primary limitation of the study. Generalizability of findings to other KAT clinics is limited. Please see text included regarding this issue in the Strengths and Limitations section. On a broader scale, some of the primary issues raised by therapists (i.e., limited access to ketamine due to legal and financial constraints), even in this restricted sample, are arguably experienced by providers across North American clinics. Ketamine remains a restricted substance in Canada and the USA which cannot be accessed without prescription for a mental illness, and therapeutic administration is costly. 

Overall, our manuscript presents one of the first studies of therapist perspectives in psychedelic therapy and the first specific to KAT, making it an important contribution to the literature.

---

## [Decision Letter · Decision Letter 1]

18 Jun 2024

“This is you teaching you:” Exploring providers’ perspectives on experiential learning and enhancing patient safety and outcomes in ketamine-assisted therapy

PONE-D-23-41326R1

Dear Dr. Argento,

We’re pleased to inform you that your manuscript has been judged scientifically suitable for publication and will be formally accepted for publication once it meets all outstanding technical requirements.

Kind regards,

Andrea Mastinu

Academic Editor

PLOS ONE

Additional Editor Comments (optional):

Reviewers' comments:

Reviewer's Responses to Questions

**Comments to the Author**

1. If the authors have adequately addressed your comments raised in a previous round of review and you feel that this manuscript is now acceptable for publication, you may indicate that here to bypass the “Comments to the Author” section, enter your conflict of interest statement in the “Confidential to Editor” section, and submit your "Accept" recommendation.

Reviewer #1: All comments have been addressed

2. Is the manuscript technically sound, and do the data support the conclusions?

Reviewer #1: Yes

3. Has the statistical analysis been performed appropriately and rigorously? 

Reviewer #1: Yes

4. Have the authors made all data underlying the findings in their manuscript fully available?

Reviewer #1: Yes

5. Is the manuscript presented in an intelligible fashion and written in standard English?

Reviewer #1: Yes

6. Review Comments to the Author

Reviewer #1: (No Response)

7. PLOS authors have the option to publish the peer review history of their article (what does this mean?). If published, this will include your full peer review and any attached files.

Reviewer #1: No

---

## [Editor Report · Acceptance letter]

20 Aug 2024

PONE-D-23-41326R1 

PLOS ONE

Dear Dr. Argento, 

I'm pleased to inform you that your manuscript has been deemed suitable for publication in PLOS ONE. Congratulations! Your manuscript is now being handed over to our production team.

Kind regards, 

on behalf of

Dr. Andrea Mastinu 

Academic Editor

PLOS ONE